# Application of Inverse Design Approaches to the Discovery of Nonlinear Optical Switches

**DOI:** 10.3390/molecules28217371

**Published:** 2023-10-31

**Authors:** Eline Desmedt, Léa Serrano Gimenez, Freija De Vleeschouwer, Mercedes Alonso

**Affiliations:** Eenheid Algemene Chemie (ALGC), Vrije Universiteit Brussel (VUB), Pleinlaan 2, 1050 Brussels, Belgium; eline.louise.desmedt@vub.be (E.D.); lea.serrano.gimenez@vub.be (L.S.G.)

**Keywords:** chemical compound space, inverse design, multistate switches, nonlinear optical properties, structure–property relationships

## Abstract

Molecular switches, in which a stimulus induces a large and reversible change in molecular properties, are of significant interest in the domain of photonics. Due to their commutable redox states with distinct nonlinear optical (NLO) properties, hexaphyrins have emerged as a novel platform for multistate switches in nanoelectronics. In this study, we employ an inverse design algorithm to find functionalized **26R**→**28R** redox switches with maximal $\beta_{HRS}$ contrast. We focus on the role of core modifications, since a synergistic effect with *meso*-substitutions was recently found for the **30R**-based switch. In contrast to these findings, the inverse design optima and subsequent database analysis of **26R**-based switches confirm that core modifications are generally not favored when high NLO contrasts are targeted. Moreover, while push–pull combinations enhance the NLO contrast for both redox switches, they prefer a different arrangement in terms of electron-donating and electron-withdrawing functional groups. Finally, we aim at designing a three-state **26R**→**28R**→ **30R** switch with a similar NLO response for both ON states. Even though our best-performing three-state switch follows the design rules of the **30R**-based component, our chemical compound space plots show that well-performing three-state switches can be found in regions shared by high-responsive **26R** and **30R** structures.

## 1. Introduction

Molecular switches are essential components of artificial molecular machines and responsive materials [1,2,3]. A molecular switch consists of a single molecule that can shift controllably between two or more stable states triggered by an external stimulus [4]. These smart switching molecules help to engineer new functionalities unattainable with silicon-based technology, leading to a broad range of applications in biomedicine [5], catalysis [6], sensors [7], and single-molecule electronics [8]. Within this context, molecules displaying distinct responses towards multiple external stimuli as well as multistate molecular switches are highly desirable as different switchable functions can be integrated into a single molecule [9,10,11].

In the field of nonlinear optics, molecules with switchable nonlinear optical (NLO) properties are key building blocks for optoelectronic and photonic devices, like high-density optical memories or logic gates [12,13,14,15]. In these dynamic systems, different NLO properties can reversibly be turned “ON” and “OFF”, such as second-harmonic generation (SHG), two-photon absorption (TPA), and third-harmonic generation (THG). At the molecular level, the NLO quantities of interest are the first (β) and second hyperpolarizability (γ), while the key factor governing the efficiency of NLO switches is the contrast between the NLO properties of the different states [12]. The amplitude of the NLO contrast can be fine-tuned by chemical design, i.e., varying the donor/acceptor substituents, the length and planarity of the π-conjugated spacer and the diradical character [16].

The quest for novel NLO materials usually relies on complementary experimental and theoretical approaches [12]. Quantum chemistry tools are very helpful to understand the structural and electronic factors governing the magnitude of the molecular hyperpolarizabilities, reducing the trial-and-error approach in the design of efficient NLO switches [17]. Most theoretical studies involve direct design approaches, in which ad hoc variation of a chemical structure is followed by evaluation of its NLO properties [18,19]. Although direct design has led to the identification of NLO switches with high first hyperpolarizability contrasts, alternative design approaches are needed for a larger-scale exploration of chemical compound space (CCS) given its size and complexity [20]. Inverse design is an innovative approach for exploring the CCS, involving the optimization of the property of interest as a function of the chemical structure. Hence, inverse design inverts the design protocol by starting with the desired properties and producing a molecule that satisfies them as an output (Figure 1A) [21]. The main advantage of all inverse design strategies is the fact that only a small fraction of a predefined CCS needs to be explored to identify new and largely improved compounds [22]. Recent advances in computational inverse design approaches have started to reshape the landscape of structures with enhanced optical functionalities available to nanophotonics [23,24].

Motivated by these findings and building on our expertise in developing and applying property optimization algorithms [22,25,26], we recently extended inverse design to the prominent field of molecular switches for nonlinear optical applications. In our long-standing interest in the design of porphyrinoid-based molecular switches [27,28,29], we selected hexaphyrin macrocycles as a test bed to assess the potential of inverse design approaches to identify hexaphyrin-based switches with enhanced NLO contrasts. Previous experimental and theoretical work demonstrated that molecular switches based on expanded porphyrins have a multiaddressable and multistate character, leading to exciting nanoelectronic applications such as conductance and optical switching [28,30,31]. On the one hand, hexaphyrins are flexible enough to switch between different π-conjugation topologies, each with distinct electronic properties and aromaticity [32]. On the other hand, hexaphyrins exhibit reversible redox chemistry providing congeneric macrocycles with [4n] and [4n + 2] π-electrons. Such redox and topology switches can be triggered by different external stimuli (Figure 1B) [33,34,35]. Furthermore, the NLO properties are easily tuneable in these macrocycles due to their structural versatility with respect to chemical functionalizations such as peripheral substitution or core modifications [28,31].

In previous studies, we applied inverse design to rationally modify redox (**26R** →**28R**, **30R**→**28R**) and topological hexaphyrin-based switches (**28M**→ **28R**) with the help of the best-first search (BFS) algorithm (vide infra) in search for well-performing NLO switches [36,37]. We succeeded in enhancing the NLO efficiency by at least an order of magnitude and identified some key players influencing the NLO contrast, i.e., the electronic character of *meso*-substituents and the centrosymmetry of the OFF state. For the **30R**-based switch, core modifications of the macrocyle were also considered, and a synergistic interplay between *meso*-substituents and core modifications was unveiled. Even though the inclusion of both types of functionalizations unlocks design potential, it also makes the prediction of the NLO contrast of hexaphyrin-based switches highly complicated and reinforces the need for innovative approaches beyond direct design.

Given the importance of core modifications for **30R**-based NLO switches [37], in this study, we revisit the **26R** →**28R** redox switch with the aim to explore the effect of core modifications on its NLO efficiency [36]. To this end, BFS optimizations of the NLO contrast are set up for the substitution pattern termed **NH_Y_R${}_{1,4}$_R${}_{2,5}$_R${}_{3,6}$** (Figure 2), with three sets of *meso*-substitution sites and one set of core modification sites combined with a diverse functionalization library. Our objective is to establish general structure–property relationships for the **26R**-based switch and compare them to the design rules devised for the **30R**→**28R** molecular switch [37]. Due to the increasing importance of multifunctional NLO materials, for the first time, we search for the optimal functionalization of a three-state molecular switch (**26R**→**28R**→**30R**) using a newly introduced figure of merit. Finally, CCS plots of the three-level switch database help us to further understand the role that each individual switch plays in the multistate switch.

## 2. Methodology

### 2.1. Inverse Design Algorithm-Best-First Search

When exploring the combinatorial chemical compound space (CCS) aiming at finding the optimal functionalization of a known molecular framework, several optimization algorithms can be employed. One of them is the greedy best-first search (BFS) approach [38,39,40], a discrete heuristic search method that optimizes the target property on a site-by-site basis assuming that the sites can be independently optimized. The discrete aspect of BFS translates to the calculation of chemically realistic molecules, i.e., functionalized derivatives of the scaffold of interest, when following the steepest gradient towards the optimum. Taking the azoheteroarene scaffold as an example of a photoswitch with 4 (sets of) sites, R${}_{1}$–R${}_{4}$, available for modification and 5 possible substituents, depicted as colored circles (Figure 3), it leads to a chemical space region of 625 potential compounds. After generating a random start structure, the first site (R${}_{1}$) is visited and 5 possible structures are constructed in accordance with the fragment library size. For each structure, the target property is evaluated and its value serves as input for the determination of the largest gradient among the 5 available substituents. As BFS assumes that the largest gradient points to the most promising optimization pathway, the corresponding functionalization is implemented, and the next site can be optimized. When all sites are iterated, it is checked whether the input and output structures are identical. If yes, the BFS procedure terminates as convergence is reached; if not, the procedure reprises with the output structure as new input. This reiteration circumvents the initial presumption of the independent-site approximation and partially ensures that the effect of neighbouring sites is accounted for.

As this optimization method is quite greedy, optima are found highly efficiently. However, depending on the roughness of the property landscape, these optima are often local, whereas one generally targets the global optimum. More effective optimization algorithms have been designed for that purpose, such as the genetic algorithm [41,42], particle swarm optimization [43], and Bayesian optimization [44,45]. Nevertheless, also for BFS, the issue of efficacy can be (partially) remedied. In this work, multiple BFS runs are conducted variying the starting structure and the site sequence. By selecting starting structures from different regions of CCS, a large enough portion of CCS is being explored. Despite the shortcoming of becoming trapped in local optima, BFS has the great advantage to search CCS locally at each iteration step by only changing one site at a time (Figure 3). The wealth of consistent structure–property information undeniably aids in deriving general design rules or guiding principles, although multiple BFS runs are advisable to gather more data.

### 2.2. Figure of Merit for the Design of NLO Switches

The first hyperpolarizability related to the Hyper-Rayleigh Scattering ($\beta_{HRS}$) is an experimentally measurable property to probe the second-order NLO responses of the different states of the switches [12,46,47,48]. This property is related to the intensity of the incoherent scattered light at twice the frequency of the incident light (2ω). Under the condition that the laser’s propagation is perpendicular to the incoherent scattered light, the HRS equations based on the full tensor components are written as Equation (Equation 1). The full tensor components can be found in Appendix A.
(1)$$\beta_{HRS}\left(-2\omega ;\omega ,\omega \right)=\sqrt{\langle \beta_{ZZZ}^{2}\rangle +\langle \beta_{ZXX}^{2}\rangle }.$$

To estimate the performance of a molecular NLO switch, a figure of merit is required. Most publications rely on the ratio between the $\beta_{HRS}$ responses of the ON and OFF states of the molecular switch to estimate the switching performance (Equation (Equation 2)) [12,28].
(2)$$\begin{array}{c} contrast=\frac{\beta_{HRS}\left(ON\right)}{\beta_{HRS}\left(OFF\right)}. \end{array}$$

Nonetheless, this ratio-based contrast definition is unsuited when the OFF state possesses $C_{i}$ symmetry, as happens in certain [28]-hexaphyrins. In this case, the denominator becomes zero and the contrast value approaches infinity. In our previous study, we circumvented this problem within the BFS procedure by assigning an arbitrary value of 0.001 a.u. to the $\beta_{HRS}$(OFF) response when the $\beta_{HRS}$(OFF) value is below 10 a.u. [36]. This arbitrary OFF-state value in case of (nearly) centrosymmetric structures has a significant impact on the NLO contrast of such hexaphyrin-based switches, as illustrated in Figure 4A. As desired, NLO switches with a centrosymmetric OFF state ($\beta_{HRS}$(OFF) ≜ 0.001 a.u. for most blue-colored marks), like Switch **3**, are preferred over those with a high $\beta_{HRS}$(OFF) response, such as Switch **2** indicated with an orange mark, although their variance is overestimated. However, switches with a high ON-state response and a relatively low OFF-state response, though larger than 10 a.u., are undervalued by the ratio-based definition. For example, Switch **1** has a $\beta_{HRS}$(ON) value of 1.72 × 10${}^{4}$ a.u., similar to that of System **3** (2.07 × 10${}^{4}$ a.u.). Nonetheless, even with a $\beta_{HRS}$(OFF) response below 500 a.u. (blue mark), Switch **1** will never be considered using the ratio-based contrast definition despite its potential, as it yields a contrast value as low as 87, in stark contrast to the ratio of 2.07 × 10${}^{7}$ evaluated for **3**.

A second NLO contrast definition introduced in the literature is based on the difference between the $\beta_{HRS}$ responses of the ON and OFF states, as written in Equation (Equation 3) [31].
(3)$$\begin{array}{c} contrast=\beta_{HRS}\left(ON\right)-\beta_{HRS}\left(OFF\right). \end{array}$$

Similar to the ratio-based contrast, a high difference value corresponds to a switch with a high $\beta_{HRS}$(ON) and a low $\beta_{HRS}$(OFF) response. As illustrated in Figure 4B, **1** is now considered as one of the best performing switches. Thus, the difference-based definition overcomes the issue of the ratio-based contrast overpromoting centrosymmetric OFF states. However, structures with a significant $\beta_{HRS}$(OFF) response may also be favored, such as Switch **2** with a $\beta_{HRS}$(OFF) value of 2.13 × 10${}^{3}$ a.u., which conflicts with the more commonly used ratio-based metric in Figure 4A.

To overcome the downsides of the previous ratio- and difference-based contrast definitions, we introduced a new metric to assess the performance of the NLO switches in our most recent work [37]. This definition (Equation (Equation 4)) steers our search of well-performing NLO switches towards a combination of high $\beta_{HRS}$(ON) states and low $\beta_{HRS}$(OFF) states and simultaneously disfavors switches with excessively high $\beta_{HRS}$(OFF) responses.
(4)$$\begin{array}{c} contrast=\frac{{\left(\beta_{HRS}\left(ON\right)-\beta_{HRS}\left(ON\right)\right)}^{2}}{\beta_{HRS}\left(ON\right)+\beta_{HRS}\left(OFF\right)}. \end{array}$$

The performance of our new figure of merit for the design of NLO switches is verified in Figure 4C. Systems **1** and **3** still remain one of the overall best-performing structures, while the NLO contrast of **2** is decreased compared to the difference-based metric. Nevertheless, it is important to note that molecular switches characterized by an enhanced ON state and a centrosymmetric OFF state like System **3** will always be dominant, irrespective of the NLO contrast metric used.

In the Results and Discussion section, the revised contrast ratio definition (Equation (Equation 3)) is applied throughout.

## 3. Results and Discussion

### 3.1. Functionalized [26]- and [30]-Hexaphyrin-Based Switches: What Sets Them Apart?

#### 3.1.1. Applying the BFS algorithm on the
**26R** → **28R** and **30R** → **28R**

In this section and for the first time, the role of core modifications, in combination with *meso*-substitution, is assessed towards the improvement of **26R** →**28R** switches based on $\beta_{HRS}$ response. An important measure for the BFS efficacy is the optimum’s improvement over the parent, namely the unsubstituted switch with a $\beta_{HRS}$ contrast of 2.09 × 10${}^{3}$ a.u. for the **26R**-based switches and 2.18 × 10${}^{3}$ a.u. for the **30R**-based switches. Figure 5 summarizes two BFS inverse design procedures, yielding the best **26R**→**28R** (A) and **30R**→**28R** (B) NLO switches discovered over all generated BFS procedures. We focus on the patterns shown in Figure 5. In the following sections, the **26R**→**28R** functionalization patterns are denoted as **NH_Y_R${}_{1,4}$_R${}_{2,5}$_R${}_{3,6}$**, and the **30R**→**28R** patterns as **X_Y_R${}_{1,4}$_R${}_{2,5}$_R${}_{3,6}$**, in accordance with **28R** functionalized structures. We note that an additional set of modifiable core modification sites (X) is included for the **30R**→**28R** switch with respect to **26R**→**28R**, since O/S/Se core modifications on X would lead to charged **26R** structures. Both procedures follow the same site iteration sequence and start from similar starting configurations: **NH_O_NH${}_{2}$_F_OH** and **S_O_NH${}_{2}$_F_OH** for **26R**→**28R** and **30R**→**28R** switches, respectively. Each site optimization step represents the introduction of a functionalization on or in the macrocycle. Strong EWGs (CN and NO${}_{2}$) are marked in red, strong EDGs (OH and NH${}_{2}$) in green, and unmodified core sites (NH) in blue; all other site functionalizations are colored in black. Individual $\beta_{HRS}$ responses and the associated contrasts of the generated structures during both BFS procedures can be found in Appendix A.

An elaborate analysis of the BFS optimization path is included for the **26R**→**28R** switch. For a detailed description of the **30R**→**28R**’s BFS run (Figure 5B), we refer to our recent publication [37]. The algorithm starts off with substituting the R${}_{3}$/R${}_{6}$
*meso*-positions of **26R**→**28R** (Figure 5A). All OFF states of the switches generated within this site iteration are centrosymmetric. Interestingly, all *meso*-substitutions enhance the NLO response of the ON state compared to the parent **26R**→**28R** switch. Nonetheless, there is a clear preference for strong EDGs, yielding an NLO contrast almost three-fold that of the EWG-substituted switches and even 10 times higher than the parent switch. The next sites up for modification are R${}_{1}$/R${}_{4}$. These *meso*-substitution positions behave similarly to the previous site iteration, where EDGs outperform EWGs. On both site combinations, R${}_{1}$/R${}_{4}$ and R${}_{3}$/R${}_{6}$, NH${}_{2}$ is favored the most.

The effect of core modifications on the NLO contrast is evaluated during the third site optimization. As all OFF states are, again, centrosymmetric, the NLO contrast mainly depends on the $\beta_{HRS}$(ON) response. Thiophene and selenophene rings behave similarly to pyrrole rings regarding the NLO contrast, with pyrrole rings ranked first. Only furan rings underperform to the other possible modifications, and thus O becomes replaced by NH on core modification sites Y. The change to NH is a crucial factor in enhancing the NLO contrast as it increases by almost 10 × 10${}^{4}$ a.u.

The final site iteration of the first global iteration consists of *meso*-substitution of sites R${}_{2}$/R${}_{5}$, where only the NH${}_{2}$ substituent results in a noncentrosymmetric OFF state (cf. green mark with lowest NLO contrast value). Remarkably, on R${}_{2}$/R${}_{5}$ positions, strong EWGs are essential for further maximizing the performance of the switch, in contrast to the other *meso*-substitution sites. The best-performing molecular switch has the **NH_NH_NH${}_{2}$_CN_NH${}_{2}$** pattern, which is a push–pull configuration containing two sets of strong EDGs and one set of strong EWGs, and without core modifications present. The NLO contrast of 5.99 × 10${}^{4}$ a.u. is 30 times larger than that of the fully unsubstituted switch. As observed from the wide ranges in NLO contrast values during the entire second global iteration, exchanging the optimal *meso*-substitution for other substituents immensely impacts the contrast, but for a handful of functionalizations. The resulting optimal switch is in line with our earlier study on the **26R**→ **28R** switch design [36].

When comparing the BFS optimization paths for the **26R** →**28R** and **30R**→**28R** switches (Figure 5A,B), we observe a different preference for certain modifications despite the structural similarity of both switches. Whereas strong EDGs were preferred on the R${}_{3}$/R${}_{6}$
*meso*-positions throughout the entire **26R**→**28R** optimization, EWGs are promoted in these positions for **30R**→**28R**. Moreover, the core modification sites Y favour any type of core modification over none. Besides their differences, both redox switches also show similarities. R${}_{1}$/R${}_{4}$ and R${}_{2}$/R${}_{5}$ share the same preference for the type of *meso*-substituents as the **26R**→**28R** switch, with few exceptions that can be attributed to the presence of a noncentrosymmetric OFF states or an unexpectedly low NLO response of the ON state. Finally, the type of core modification on sites X, only considered for **30R**→**28R**, inclines more towards the same preference as the sites Y of the **26R**→**28R**. This trend, however, collapses during the last two global iterations as inserting the same core modifications on sites X and Y, more specifically two sets of oxygen atoms, further boosts the NLO contrast to a value of 2.08 × 10${}^{4}$ a.u.

In conclusion, the best-performing patterns resulting from the BFS procedures are **NH_NH_NH${}_{2}$_CN_NH${}_{2}$** for the **26R**-based NLO switch and **O_O_NH${}_{2}$_CN_CN** for the **30R**-based switch. We note that three additional BFS runs using start structures for which a variation in type of core modifiers was imposed resulted in our aforementioned optimal structure. This makes it very plausible that we identified one of the most NLO-efficient **26R**-based switches. The **26R** and **30R** optima confirm our hypothesis that the preferred number of EWGs and EDGs on the macrocycle depends on the oxidation state of the redox switch. Furthermore, the **30R**-based switches favour the incorporation of core modifications, contrary to the **26R**-based switch. Nevertheless, both optima are characterized by an enhanced ON state and a centrosymmetric OFF state, in line with our previous studies [36,37]. Through functionalization, the total NLO contrast can be enhanced by a factor ranging from 10 for **30R**-based switches to even 30 for **26R**-based switches. To assess the role of each functionalization towards the total NLO response, a steepest ascent approach is applied on the two optimal switches.

#### 3.1.2. Steepest Ascent of the BFS Optima

The steepest ascent methods start from the unsubstituted molecular switch and gradually builds up towards the optimum, following the largest property gradient. The number of steps is equal to the number of functionalized sites. For the **26R**- and **30R**-based BFS optima, this number is three and five, respectively. Table 1 and Figure 6 summarize our analysis of the steepest ascent results, in which the NLO response of the optima is decomposed with respect to the functionalization’s position and type. Our main figure of interest is the %(B-P), which quantifies how much each modification contributes to the overall improvement of the best-performing switch over the parent switch.

In the first step, all possible functionalizations enhance the NLO contrast with respect to the parent structure. The EDGs on R${}_{1,4}$ and R${}_{3,6}$ each contribute around 19% and 15%, respectively, to the best NLO contrast, while the EWG on R${}_{2,5}$ only yields a 6% improvement. Although similar contributions are observed for the two sets of EDGs, NH${}_{2}$ substituents on R${}_{1,4}$ perform slightly better. In Step 2, we observe that adding a second set of NH${}_{2}$ substituents on R${}_{1,4}$ leads to the largest enhancement. The combination of strong EDGs on R${}_{1,4}$ and R${}_{3,6}$ is therefore preferred over a push–pull pattern, in which NH${}_{2}$ on R${}_{3,6}$ is combined with CN on R${}_{2,5}$, although the differences are small (5%). The strong EDGs on R${}_{3,6}$ provide a 26% improvement, i.e., 11% larger than their contribution in Step 1. In other words, the *meso*-substitutions combined act synergistically, with the most dramatic effect for the push–pull-type switch (+15%).

Remarkably, the two EDGs placed on their respective positions do not even account for half of the total NLO contrast. Consequently, the EWGs on R${}_{2,5}$, finally introducing a push–pull substitution pattern, are responsible for a 55% enhancement to the maximal NLO contrast. We again take note that these CN groups only become so decisive for the NLO contrast due to the presence of the other NH${}_{2}$ substituents. Indeed, only including CN groups on the R${}_{2,5}$ hardly enhances the NLO contrast compared to the unsubstituted structure, but in combination with one or particularly two sets of NH${}_{2}$, its full potential is shown via a synergistic effect.

For the **30R**-based molecular switch, the main findings for the steepest ascent to the optimal pattern, **O_O_NH${}_{2}$_CN_CN**, are summarized in Figure 6B. Here, all *meso*-substituents only account for 50% of the total NLO contrast. First, inclusion of the strongly electron-withdrawing cyano groups on R${}_{3,6}$ is favored with an increase of 14% with respect to the parent. The NH${}_{2}$ groups on R${}_{1,4}$ are selected as the next modification, establishing a push–pull effect, but they similarly only account for an additional 14% of the maximal NLO contrast improvement compared to the parent switch. The third set of *meso*-substituents (CN on R${}_{2,5}$) is responsible for a 23% further enhancement of the NLO contrast. The last functionalizations correspond to the core modifications with oxygen on sites Y and X, contributing 29% and 21%, respectively. Thus, core modifications entail a 50% increase in the NLO contrast. Due to the combination of core modifications and *meso*-substitutions, the NLO contrast is synergistically improved as shown by the steepest ascent analysis.

In summary, from our inverse design procedures, two possible functionalization combinations emerge that can synergistically enhance the NLO contrast. The first combination, the push–pull effect, is beneficial for both types of redox switches and consists of the integration of strong EDGs coupled to strong EWGs. The second functionalization combination encompasses the interplay between core modifications and *meso*-substitutions, which can synergistically amplify the NLO contrast in case of **30R** → **28R**.

#### 3.1.3. Dataset Comparison: **26R** → **28R** versus **30R** → **28R**

In previous sections, we established that the two redox switches have their own ideal recipe to maximize the NLO contrast. In an attempt to generalize our findings, an analysis of the generated **26R** →**28R** database is carried out to derive guiding principles for the design of high-potential **26R**-based NLO switches, and these findings are directly compared to our previously established design rules for **30R**→**28R** [37]. For the details, we refer to Appendix A. From our previous studies [36,37] and the BFS runs performed in this work, we collected 242 and 320 different functionalization patterns for the **26R**→**28R** and **30R**→ **28R** switches, respectively. Figure 7 summarizes our main findings.

It is clear that much higher NLO contrasts can be achieved for the **26R**-based switch than for the **30R** switch, with the majority of contrast values below 1.50 × 10${}^{4}$ a.u. for the former and 8.00 × 10${}^{3}$ a.u. for the latter. However, for both the **26R** and **30R**-based switches, only select sets of 8% and 4% have contrasts larger than 3.00 × 10${}^{4}$ a.u. and 1.60 × 10${}^{4}$ a.u., respectively. Of course, the two BFS optima retrieved in Section 3.1.1 fall within these high-NLO contrast groups.

Since our interest lies in obtaining design rules for the best-performing hexaphyrin-based switches, we redistributed our two datasets into four groups based on their respective quartiles. Our target group is primarily Q3–Q4, or the best 25%, as this group contains the patterns with the highest NLO contrasts. We examine, for both hexaphyrin redox switches, which core modifications and *meso*-substitutions are the most populated in the different groups.

First, we remark again that no structures with core modifications on sites X are generated for the **26R** →**28R**, because this site combination would lead to charged ON states. In general, we observe that in our databases, structures with core modifications exceed the number of all-aza structures for both switches (Figure 7B). Nevertheless, for the **26R**→**28R** switches, most patterns without core modifications are positioned in the 50% best-performing NLO switches (Q2–Q3 and Q3–Q4). By contrast, only 6.25% of the best-performing **30R**→ **28R** structures in Q2–Q3 and Q3–Q4 are not core modified (Figure 7B), and the number of all-aza structures increases, moving to lower NLO contrasts. As witnessed for the 50% best-performing **30R**-based switches in Figure 7B, there is a preference to modify only sites Y, with the presence of core modifications on sites X being less favoured.

Next, we focus on the *meso*-substitutions on R${}_{1,4}$, R${}_{2,5}$, and R${}_{3,6}$. Figure 7C displays the number of structures containing a combination of zero, one, two or three pairs of EDGs and/or EWGs in the top and bottom quartile for both types of switches. In the heatmaps, only strong EDGs and strong EWGs are counted in, so F and CH${}_{3}$ are left out. For the **26R**-based switch, combinations of one or two pairs of EDGs together with one set of EWGs are mostly present in the top 25%. The preference for more EDGs than EWGs is also witnessed from the last column, containing 16 structures, or around 25% of the group size, with a minimum of two pairs of EDGs but no EWGs. On the contrary, the majority of the worst-performing switches in group Q0-Q1 have a *meso*-substitution pattern with only EWGs and no EDGs. Hence, we can conclude that the presence of at least one set of EDGs is crucial in enhancing the NLO contrasts of **26R**-based switches. For the **30R**-based switches, the Q3–Q4 group consists of combinations of EWGs and EDGs, but here the largest percentage of structures contains two sets of EWGs and one set of EDGs, while such combinations are less populated within the best-performing **26R** → **28R** switches. Note that each switch in the top 25% contains at least one set of EWGs. Three patterns dominate in the low-contrast Q0–Q1 group: patterns with only EDGs, only EWGs and no EWGs/EDGs. Therefore, both hexaphyrin-based NLO switches benefit from a combination of EDGs and EWGs (push–pull effect) to increase the NLO contrast. Nevertheless, the NLO contrast of the [26]-hexaphyrins is also enhanced with only EDGs, while this substitution pattern significantly decreases the NLO contrast of [30]-hexaphyrins.

In summary, two different design rules emerged from the statistical analysis of the databases generated for each redox switch. First, the **26R** →**28R** molecular switch prefers more EDGs than EWGs on their macrocycle, although also fully EDG patterns can enhance the NLO contrast. Second, the **30R**→ **28R** molecular switch prefers the opposite peripheral substitution pattern with more EWGs than EDGs.

### 3.2. Towards Optimal Hexaphyrin Based Three-State Molecular Switches

Due to the ability of the redox states to interconvert between each other through two-electron redox reactions [49], a multistate switch can be constructed based on two ON states (**26R** and **30R**) and a single OFF state (**28R**). The following question arises: Which of the established design rules devised for the individual switches prevails during the optimization of the three-state molecular switch? In this section, we perform two BFS optimizations from different starting structures to improve the performance of the three-state NLO switch (**26R** →**28R**→**30R**), with the performance defined as the comparable enhancement of the NLO contrast of both ends of the three-state switch, i.e., the **26R**→**28R** and **30R**→**28R** switches.

#### 3.2.1. Finding the Best Three-State Hexaphyrin Switch with the BFS Algorithm

To evaluate the efficiency of the three-state switch, a new target function must be introduced to be optimized using inverse design. Our goal is to keep the $\beta_{HRS}$(OFF) response (**28R**) as low as possible, while maximizing the NLO response of the two ON states (**26R** and **30R**). Furthermore, we aim for a figure of merit that increases the NLO contrasts of both individual switches to a similar value. Hence, we propose Equation (Equation 5) as our new metric, with an explicit dependence on the contrast values of the **26R**→**28R** and **30R**→ **28R** switches.
(5)$$function=\left\{\begin{array}{cc} \frac{contrast(26R\to 28R)+contrast(30R\to 28R)}{2\times \frac{\beta_{HRS}\left(26R\right)}{\beta_{HRS}\left(30R\right)}}, & \mathrm{if}\;\beta_{HRS}\left(26R\right)>\beta_{HRS}\left(30R\right)\\ \frac{contrast(26R\to 28R)+contrast(30R\to 28R)}{2\times \frac{\beta_{HRS}\left(30R\right)}{\beta_{HRS}\left(26R\right)}}, & \mathrm{if}\;\beta_{HRS}\left(26R\right)<\beta_{HRS}\left(30R\right)\\ \frac{contrast(26R\to 28R)+contrast(30R\to 28R)}{2},\; & \mathrm{if}\;\beta_{HRS}\left(26R\right)=\beta_{HRS}\left(30R\right) \end{array}\right..$$

This metric penalises the average of the NLO contrast of both switches according to the dissimilarity between the $\beta_{HRS}$(ON) responses of **26R** and **30R** states defined as the ratio between their $\beta_{HRS}$ values. In other words, if both ON-state responses are more or less equivalent, the function is approximately equal to the average of the switch contrasts. However, if the ON-state values differ from each other by a significant factor, the contrast average is reduced by the same factor (>1).

Figure 8 summarizes the BFS runs by displaying the optimization steps with respect to the NLO function (A), the **26R**→**28R** NLO contrast (B) and the **30R**→**28R** NLO contrast (C). We select the unsubstituted **26R**→**28R**→**30R** as starting structure of this BFS run, for which four sites (Y, R${}_{1,4}$, R${}_{2,5}$ and R${}_{3,6}$) are available for functionalization. Tables with the detailed BFS results for the different global iterations can be found in Appendix A.

The first sites to be optimized are the *meso*-positions R${}_{2}$/R${}_{5}$, where the maximum function value (3.25 × 10${}^{3}$ a.u.) is obtained for the CN group. The **28R** OFF state is centrosymmetric and both the **26R** and **30R** states exhibit the highest $\beta_{HRS}$(ON) values with this strong EWG, ensuring a maximal NLO contrast value for both redox switches. Next, the NH${}_{2}$ group is incorporated on the R${}_{3}$/R${}_{6}$
*meso*-positions creating a push–pull pattern with a function value of 4.31 × 10${}^{3}$ a.u. The OFF state remains centrosymmetric and the **26R**-based NLO contrast is the highest among all possible substitutions, while the **30R**-based contrast is the second best, after the CN-substituted one. However, as CN performs the worst for **26R**→**28R**, the resulting function value is half as low. This example nicely highlights the clear difference in preferred functionalization between the redox states. Substituting *meso*-positions R${}_{1}$/R${}_{4}$ results in all OFF states being centrosymmetric, except for NO${}_{2}$. The maximal contrast value for the **26R**→**28R** switch reaches 5.99 × 10${}^{4}$ a.u. for R${}_{1}$/R${}_{4}$ = NH${}_{2}$, which corresponds to the BFS optimum for the individual **26R**→ **28R** switch (Section 3.1.1, Figure 5C). For the **30R**-based switch, the maximal NLO contrast is at 1.25 × 10${}^{4}$ a.u. containing CN instead of NH${}_{2}$ at R${}_{1}$/R${}_{4}$. Since the average NLO contrast for the **30R**→**28R** is much lower than for the **26R**→**28R**, our proposed metric depends more strongly on the contrast of the **30R**→**28R** switch. Indeed, the highest function value is attained for the **NH_NH_CN_CN_NH${}_{2}$**
*meso*-substituted pattern with a value of 9.65 × 10${}^{3}$ a.u. We note that the function value indeed becomes closer to the ON state values of both individual switches. Finally, the different core modifications are tested for this *meso*-substitution optimum. All three core modifications (O, S or Se) provide higher function values than the three-level switch without core modifications, with small differences between the various heteroatoms. The best-performing three-state switch retrieved during this first global iteration has the **NH_S_CN_CN_NH${}_{2}$** pattern with a function value of 1.12 × 10${}^{4}$ a.u. and has the highest NLO contrast for **30R**. The **26R**-based switch still prefers no core modifications, making the **30R**-based switch the decisive factor for selecting the optimal pattern of the three-state switch.

The second global iteration immediately starts with finding a new optimum with a function value of 1.50 × 10${}^{4}$ a.u. by replacing the CN group on the *meso*-positions R${}_{1}$/R${}_{4}$ by NO${}_{2}$. Even though there exist several substituents yielding a higher **26R**→**28R** contrast, the corresponding NLO contrast of **30R**→**28R** becomes significantly lower for these patterns, making them unsuitable for our metric. The remaining three site optimizations (Y, R${}_{3}$/R${}_{6}$ and R${}_{2}$/R${}_{5}$) do not improve the performance of the three-state switch. Remarkably, for these site optimizations, the highest function values align with the highest or near-highest individual **26R**- and **30R**-based NLO contrasts. Also, we observe that introducing NH or O on the sites Y results in noncentrosymmetric OFF states, significantly decreasing the individual contrasts. Additionally, we again find that EDGs on the R${}_{3}$/R${}_{6}$ positions are essential for both switches to maintain the push–pull substitution pattern. The BFS optimum for the three-state switch is thus the **NH_S_NO${}_{2}$_CN_NH${}_{2}$** pattern with a function value of 1.50 × 10${}^{4}$ a.u., which is only 1.4 times lower than the optimal **30R**-based switch but 4 times lower than the best-performing **26R**-based switch.

In our previous study on **30R** →**28R**, this pattern, **NH_S_NO${}_{2}$_CN_NH${}_{2}$**, was ranked as the sixth best performing pattern generated by the BFS procedures [37]. This observation is in parallel with our proposed function value for the performance of multistate switches being more prone towards better performing **30R**→**28R** than **26R**→**28R** due to the lower contrast values found for the former. To confirm whether an alternative pattern could be found closer to the best-performing patterns for **26R**→**28R**, we carried out an additional BFS optimization starting from the most optimal structure found in Section 3.1.1, namely **NH_NH_NH${}_{2}$_CN_NH${}_{2}$**. Convincingly, after four global iterations, the algorithm results in the same optimum as our initial BFS procedure. However, the question still remains whether a synergistic effect exists between the different types of functionalizations in the optimal switch.

#### 3.2.2. Steepest Ascent of the Best **26R** → **28R** → **30R** Molecular Switch

To answer this question, we apply the steepest ascent method (Table 2 and Figure 9) following the largest gradient in the function value to build up our optimal three-state switch.

The first build-up step shows that only CN at R${}_{2}$/R${}_{5}$ and NH${}_{2}$ at R${}_{3}$/R${}_{6}$ yield improved function values compared to the parent switch, with the former leading to the largest function value (3.25 × 10${}^{3}$ a.u.), a 9% increase over the parent. Whereas the NH${}_{2}$ group would be the favored choice for **26R**, CN at R${}_{2}$/R${}_{5}$ is preferred for **30R**. This seems to contradict the results of the steepest ascent on the **30R**→**28R** optimum. However, our previous research revealed that the exact positioning of EDGs and EWGs on the macrocyles (R${}_{1}$/R${}_{4}$ vs. R${}_{3}$/R${}_{6}$) can also substantially impact the efficiency of the switch [37]. Next, NH${}_{2}$ is positioned on R${}_{3}$/R${}_{6}$ and is responsible for an additional increase of 8% in the NLO contrast metric. We note that immediately introducing a second strong EWG on the macrocycle is disadvantageous for the performance of the three-state switch. Curiously though, this step follows the **26R** preference and not that of **30R**. In the third step, the core modification is included, but it only contributes 3%. Although the contribution of S is small, its impact mainly relies on the conservation of the centrosymmetric OFF state. Indeed, the $\beta_{HRS}$ value of **28R(NH_S_H_CN_NH${}_{2}$)** is 0 a.u., while **28R(NH_NH_NO${}_{2}$_CN_NH${}_{2}$)** has a $\beta_{HRS}$ value of 2.92 × 10${}^{3}$ a.u. Unexpectedly, the three functionalizations so far only contribute 20% to the best-performing three-state switch. That means that 80% of the multistate optimum’s enhancement is ascribed to the insertion of NO${}_{2}$ on R${}_{1}$/R${}_{4}$. Due to the presence of the S core modifier, introducing NO${}_{2}$ finally yielded a centrosymmetric OFF state, unlike the NO${}_{2}$ patterns in the previous steps. The synergistic effect of the Y core modifications manifests itself in the response of the OFF state, which is in stark contrast to our recent findings on the **30R**→**28R** switch where the core modifications were responsible for the enhancement of the $\beta_{HRS}$(ON) response [37].

#### 3.2.3. Chemical Space Visualization

Altogether, 146 patterns were retrieved for the three-state switch. As mentioned in Section 3.1.1, the NLO contrast of hexaphyrin-based switches is mainly dependent on the $\beta_{HRS}$(ON) response. Taking a closer look at the $\beta_{HRS}$ values over all ON states generated during the BFS procedures on the multistate switch (Figure 10), we remark that the largest values correspond to **26R** ON states. Only 8% of all generated switches have ON responses above 2.25 × 10${}^{4}$ a.u., and all correspond to **26R** states. The majority of **30R** structures have $\beta_{HRS}$(ON) responses in the range of 0–15,000 a.u. Hence, **26R**→**28R** is more likely to outperform the **30R**→ **28R**-based switches in terms of individual NLO contrast.

Because the BFS procedure sequentially visits all modifiable sites, a large set of structures in the vicinity of first the initial structure and later each intermediate optimum is generated, with each optimization step covering larger portions of the CCS. To investigate which regions of the space yield the most efficient three-state switches, we visualized the CCS through t-SNE plots by confining the space through the extended connectivity fingerprints (ECFP) (Figure 11). Different measures were used to color the data points of the t-SNE plots. Regions of interest were aggregated by a colored shape. We note that the t-SNE plots contain all functionalization patterns gathered in this and previous studies from our group and are therefore not restricted to the three-state switches; a complete list is included in Appendix A.

We started by investigating the regions that contain a certain number of core modifications (Figure 11A). Almost all structures with either none or two sets of core modifications were grouped together in a red oval and blue box, respectively. If we compare the CCS of the **26R** → **28R** (Figure 11B) and **30R** →**28R** switches (Figure 11C), we observe that distinct regions of the CCS are favoured depending on the type of redox switch. A large part of the structures that are not core modified are performing significantly well for **26R**-based NLO contrasts, as represented by the overlap between the red and yellow ovals. We note that there are no **26R**-based switches located in the blue rectangle, as two sets of core modification sites were not examined for this type of switch. On the other hand, for the **30R**-based NLO contrast, we found more switches with high NLO contrasts in this blue region, as indicated by a green oval within the blue rectangle in Figure 11C. Furthermore, the region with no core modifications (red oval) does not contain many structures with high NLO contrasts. Other smaller regions marked by a yellow or green oval were found outside of these blue and red regions. These structures only had one out of two core modification sites modified. Finally, Figure 11D highlights the data points according to their function value as metric for the NLO contrast of the three-state switch. Most of the top-performing three-state switches are positioned in the regions with a high **30R**→**28R** NLO contrast, and some of them reside in the intersection of the yellow and green ovals, regions delimited by a black line in Figure 11D. This demonstrates that, although maximizing the **30R**→**28R** NLO contrast is key for elevating the performance of the three-state switch, it does not imply that these switches have patterns very different from the ideal **26R**→ **28R**.

## 4. Conclusions

Building further on our previous work on the design of functionalized hexaphyrin-based molecular switches with enhanced NLO contrast, we investigate the effect of core modifications in combination with *meso*-substitutions for the **26R** →**28R** redox switch with the help of the BFS algorithm. The optimal **26R**-based switch combines two sets of EDGs and one set of EWGs, but does not include any core modifiers. In several facets, this optimal functionalization differs from the optimal pattern found for the **30R**→**28R** molecular switch [37], despite the structural similarity of both redox switches. Not only do core modifications synergistically help to enhance the contrasts of **30R**-based switches, but these switches are also characterized by a distinct push–pull character, with two pairs of EWGs combined with one pair of EDGs. A thorough analysis of our large **26R**→**28R** and **30R**→ **28R** databases confirms that the design principles devised for the BFS optima can be generalized for the overall best-performing hexaphyrin-based switches. Through a steepest ascent analysis of the optima, we also conclude that there is a synergistic effect in amplifying the ON state’s NLO response between either the *meso*-substituents of the push–pull patterns or between the core modifications and the push–pull substitution pattern.

Because the three redox states are interconvertible between each other, we additionally optimize the performance of the three-state switch **26R** →**28R**→**30R** by maximizing the response for both ON states (**26R** and **30R**) and minimizing the response of the OFF state (**28R**). Our aim is to discover which of the design rules would prevail during the BFS optimization for the multistate switch. Our best-performing three-state switch has a pattern similar to the **30R**→**28R** due to the lower NLO response of the **30R** state compared to that of **26R**. Again, a synergistic effect is responsible for the enhancement of the performance of the three-state switch, although in this case, the synergistic effect manifests itself by keeping the OFF state centrosymmetric instead of enhancing the ON state. Finally, we visualized the chemical space and identified the regions that represent enhanced **26R**→**28R** and **30R**→**28R** switches. Even though high-potential three-state switches are generally found in regions of increased **30R**-based systems, some overlap with the **26R**→ **28R** regions is seen, demonstrating the compatibility of the two types of switches despite their dissimilarities.

## Figures and Tables

**Figure 1 molecules-28-07371-f001:**
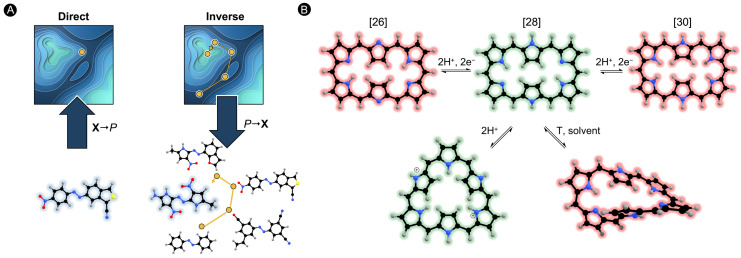
(**A**) Schematic representation of direct and inverse design approaches. Inverse design follows the reverse design protocol starting with the property of interest and ending in the optimal structure. (**B**) Different structures in hexaphyrin macrocycles interconvertible upon redox and protonation reactions as well as changing solvation and temperature.

**Figure 2 molecules-28-07371-f002:**
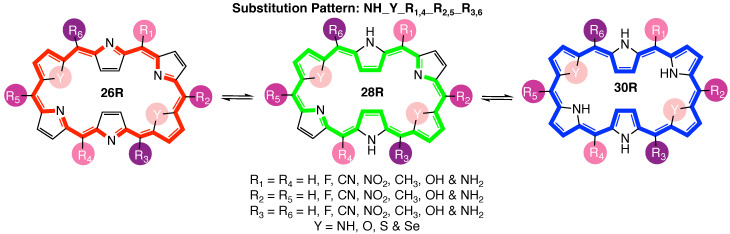
Graphical representation of the three-state hexaphyrin-based redox switch **26R** →**28R**→**30R**, indicating the three pairs of *meso*-substitution sites {R${}_{1,4}$; R${}_{2,5}$; R${}_{3,6}$}, the one set of core modification sites {Y} as well as the functionalization library.

**Figure 3 molecules-28-07371-f003:**
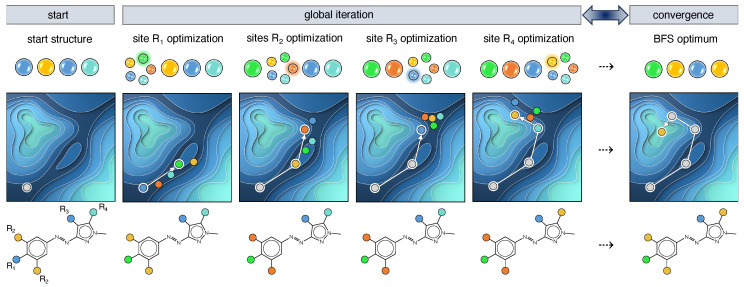
Schematic representation of the best-first search algorithm applied on an azoheteroarene photoswitch with 4 modifiable (sets of) sites R${}_{1}$–R${}_{4}$ and a fragment library of 5 substituents depicted as blue-, cyan-, green-, yellow- and orange-colored circles.

**Figure 4 molecules-28-07371-f004:**
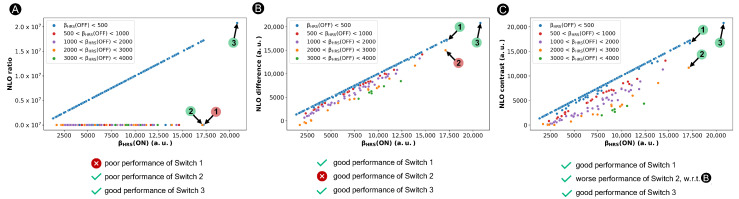
Scatter plots of three contrast definitions based on the ratio (**A**), difference (**B**) and a revised ratio (**C**) versus the $\beta_{HRS}$ of the **30R** structures. Three representative switches, Switches **1**–**3**, are highlighted. If the system is categorized correctly by the NLO contrast definition, it is labelled with a green circle, otherwise with a red circle.

**Figure 5 molecules-28-07371-f005:**
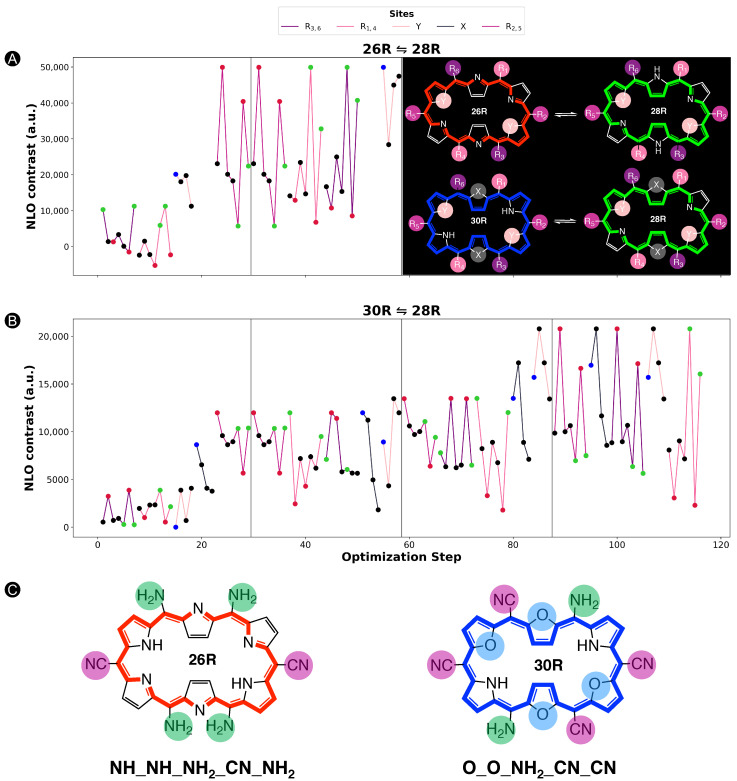
Chronological site order of the BFS procedure maximizing the NLO contrast for (**A**) **26R**→**28R** and (**B**) **30R**→**28R** switches starting from **NH_O_NH${}_{2}$_F_OH** and **S_O_NH${}_{2}$_F_OH**, respectively. Black lines separate the global iterations. The data points highlighted in blue, red and green are the sites with NH groups, strong EWGs and strong EDGs, respectively. The gaps in (**A**) correspond to the sites X that were not considered for the **26R**→ **28R** switch. The ON states of the optimal substituted molecular switches for both BFS runs are presented in (**C**) with indication of the substitution pattern.

**Figure 6 molecules-28-07371-f006:**
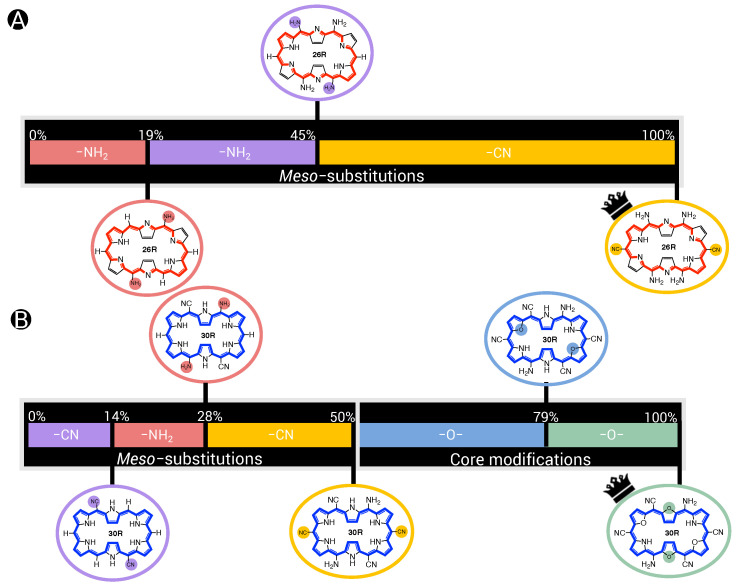
Steepest ascent of our best-performing NLO switches (**A**) **NH_NH_NH${}_{2}$_CN_NH${}_{2}$** for **26R**→**28R** and (**B**) **O_O_NH${}_{2}$_CN_CN30R**→ **28R**. The percentage represents the contribution of each step to the maximal NLO response relative to the unsubstituted structure.

**Figure 7 molecules-28-07371-f007:**
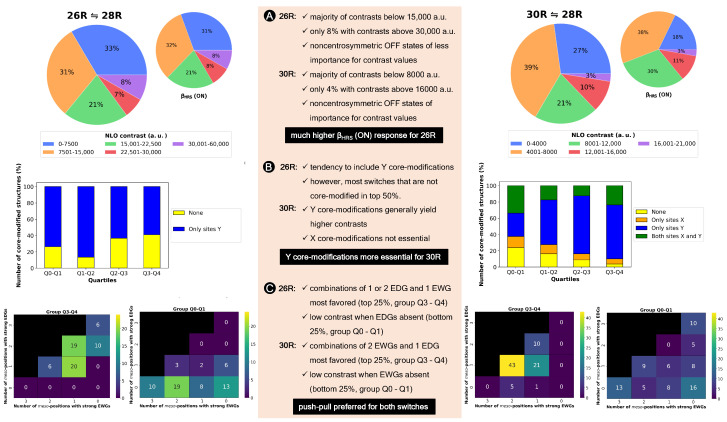
(**A**) Pie diagrams of NLO contrast and $\beta_{HRS}$(ON) of the **26R**→**28R** and the **30R**→**28R** divided into groups ranging 7.50 × 10${}^{3}$ a.u. and 4.00 × 10${}^{3}$ a.u., respectively. (**B**) Barplots of the number of core-modified structures in percentage versus quartiles for both switches. (**C**) Heatmap of the number of functionalized switches against the number of strongly electron-donating groups versus strongly electron-withdrawing groups on the *meso*-positions for the Q3–Q4 and Q0–Q1 quartiles.

**Figure 8 molecules-28-07371-f008:**
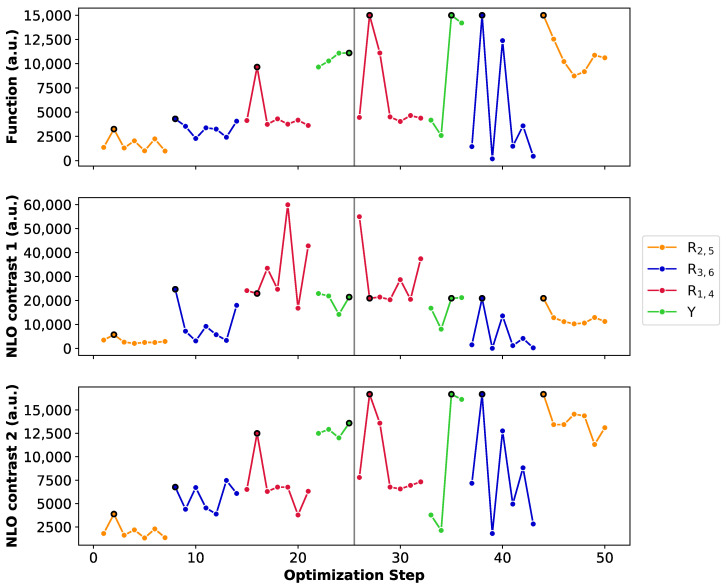
Chronological site order of the BFS procedure maximizing the function value of the **26R** →**28R**→**30R** switch starting from the unsubstituted structure. Black lines separate the global iterations. NLO contrast 1 and NLO contrast 2 stand for the individual **26R**→**28R** and **30R**→ **28R** switches, respectively. We remark the different y-axis range for the NLO contrast 1.

**Figure 9 molecules-28-07371-f009:**
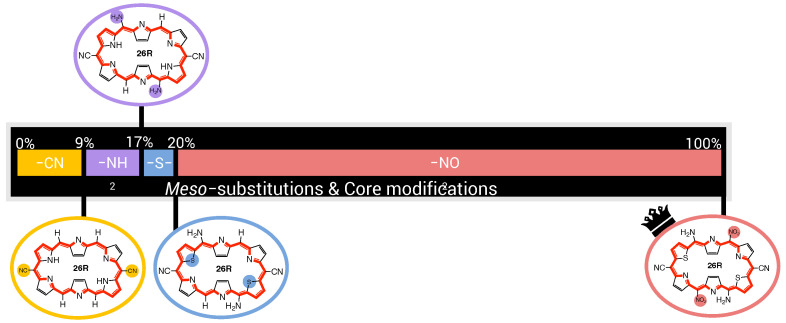
Steepest ascent of our best-performing three-state switch for NLO applications with pattern **NH_S_NO${}_{2}$_CN_NH${}_{2}$**. The percentage quantifies the contribution of each steepest ascent step to the maximal NLO response as defined by our function value relative to the unsubstituted structure.

**Figure 10 molecules-28-07371-f010:**
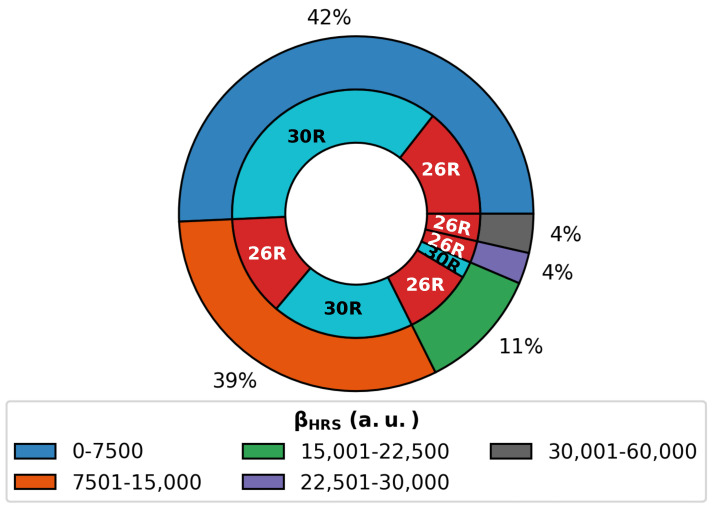
Pie diagram of $\beta_{HRS}$ of the **26R** and **30R** structures divided into groups with ranges of 7500 a.u. For each group, the number of **26R** and **30R** structures is colored in red and blue, respectively.

**Figure 11 molecules-28-07371-f011:**
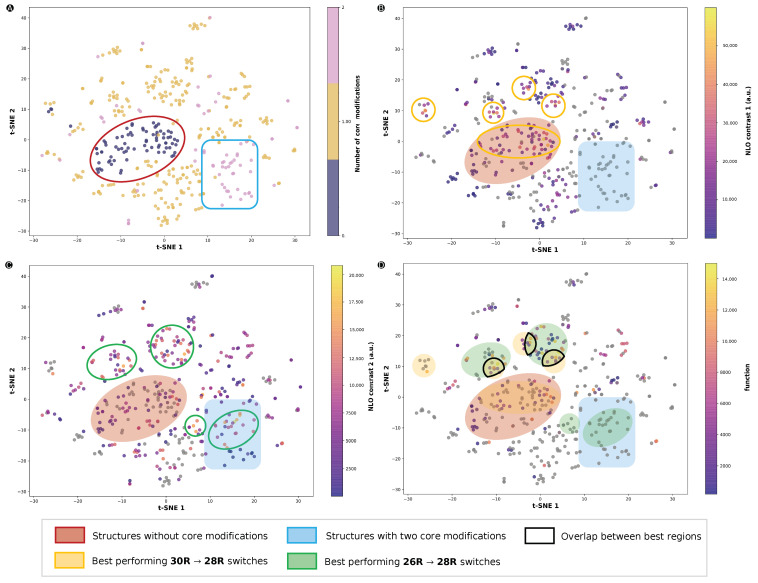
t-SNE plots of the chemical compound space described by the ECFP fingerprints with different measures: (**A**) number of core-modified sites (S, O or Se), (**B**) the **26R** →**28R** NLO contrast, (**C**) the **30R**→**28R** NLO contrast and (**D**) the function value of the three-state switches. NLO contrasts 1 and 2 refer to **26R**- and **30R**-based switches, respectively. Regions of the CCS highlighted by a red oval are structures without core modifications, while the blue rectangle represents structures with two core-modified sites, X and Y. Green and yellow ovals highlight regions with high NLO contrasts for **30R**→**28R** and **26R**→ **28R**, respectively.

**Table 1 molecules-28-07371-t001:** Steepest ascent of our best-performing **26R**-based switch: **NH_NH_NH${}_{2}$_CN_NH${}_{2}$**. %B represents the percentage in the NLO constrast of all switches in a steepest ascent step with respect to the NLO contrast maximum. %(B-P) is similar to %B but takes the parent as a reference. %(CB) stands for the percentage in the NLO constrast of all switches in a steepest ascent step with respect to the current best.

Step	X	Y	R${}_{1,4}$	R${}_{2,5}$	R${}_{3,6}$	Contrast (a.u.)	%B	%(B-P)	%(CB)
Parent	NH	NH	H	H	H	2.09 × 10${}^{3}$	3	0	-
1	**NH**	**NH**	**NH${}_{2}$**	**H**	**H**	**1.32 × 10${}^{4}$**	**22**	**19**	**100**
	NH	NH	H	CN	H	5.73 × 10${}^{3}$	10	6	33
	NH	NH	H	H	NH${}_{2}$	1.09 × 10${}^{4}$	18	15	80
2	NH	NH	NH${}_{2}$	CN	H	2.53 × 10${}^{4}$	42	40	89
	**NH**	**NH**	**NH${}_{2}$**	**H**	**NH${}_{2}$**	**2.83 × 10${}^{4}$**	**47**	**45**	**100**
3	**NH**	**NH**	**NH${}_{2}$**	**CN**	**NH${}_{2}$**	**5.99 × 10${}^{4}$**	**100**	**100**	**100**

**Table 2 molecules-28-07371-t002:** Steepest ascent of our best-performing multistate switch with pattern **NH_S_NO${}_{2}$_CN_NH${}_{2}$**. %B represents the percentage in NLO function of all switches in a steepest ascent step with respect to the NLO contrast maximum. %(B-P) is similar to %B but takes the parent as a reference. %(CB) stands for the percentage in NLO function of all switches in a steepest ascent step with respect to the current best.

Step	X	Y	R${}_{1,4}$	R${}_{2,5}$	R${}_{3,6}$	contrast${}_{26R\to 28R}$ (a.u.)	Contrast${}_{30R\to 28R}$ (a.u.)	Function (a.u.)	%B	%(B-P)	%(CB)
Parent	NH	NH	H	H	H	2.09 × 10${}^{3}$	2.18 × 10${}^{3}$	2.04 × 10${}^{3}$	14	0	-
1	NH	S	H	H	H	1.36 × 10${}^{3}$	2.01 × 10${}^{3}$	1.15 × 10${}^{3}$	8	−7	−74
	NH	NH	NO${}_{2}$	H	H	9.52 × 10${}^{0}$	3.47 × 10${}^{3}$	3.00 × 10${}^{2}$	2	−13	−144
	**NH**	**NH**	**H**	**CN**	**H**	**5.73 × 10${}^{3}$**	**3.88 × 10${}^{3}$**	**3.25 × 10${}^{3}$**	**22**	**9**	**100**
	NH	NH	H	H	NH${}_{2}$	1.94 × 10${}^{4}$	3.49 × 10${}^{3}$	2.30 × 10${}^{3}$	15	2	22
2	NH	S	H	CN	H	3.87 × 10${}^{3}$	1.35 × 10${}^{4}$	3.75 × 10${}^{3}$	25	13	75
	NH	NH	NO${}_{2}$	CN	H	1.35 × 10${}^{3}$	4.31 × 10${}^{3}$	1.57 × 10${}^{3}$	10	−4	−21
	**NH**	**NH**	**H**	**CN**	**NH${}_{2}$**	**2.47 × 10${}^{4}$**	**6.76 × 10${}^{3}$**	**4.31 × 10${}^{3}$**	**29**	**17**	**100**
3	**NH**	**S**	**H**	**CN**	**NH${}_{2}$**	**2.04 × 10${}^{4}$**	**6.94 × 10${}^{3}$**	**4.65 × 10${}^{3}$**	**31**	**20**	**100**
	NH	NH	NO${}_{2}$	CN	NH${}_{2}$	1.68 × 10${}^{4}$	3.77 × 10${}^{3}$	4.17 × 10${}^{3}$	28	16	82
4	**NH**	**S**	**NO${}_{2}$**	**CN**	**NH${}_{2}$**	2.09 × 10${}^{4}$	1.67 × 10${}^{4}$	**1.50 × 10${}^{4}$**	**100**	**100**	**100**

## Data Availability

The data will be made available upon request.

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
