# Peer review of "Application of Inverse Design Approaches to the Discovery of Nonlinear Optical Switches"

_molecules, 2023, doi:10.3390/molecules28217371_

Round 1

Reviewer 1 Report

Comments and Suggestions for Authors

Comments: The article entitled “Application of inverse design approaches to the discovery of

nonlinear optical switches” is overall a good scientific effort that has been made by Mercedes Alonso et al. The background and purpose of this study is interesting and the research article is properly handled by the authors. I believe that this article fulfills the quality to be published in Journal of molecules and can be of interest for its readership after the suggested modifications.

Recommendation: Accept with minor revision

1.      The abstract need be precise and reduced for better understanding as in the current form it is too lengthy.

2.      Keywords should be in the alphabetical order.

3.      The introduction part needs rephrasing….. and also need to be reduced

4.      Lines 132-133, “Taking the azoheteroarene scaffold as an example of a photoswitch scaffold (Figure 3) with 4 sites available for modification and 5 possible substituents, it leads to a chemical space region of 625 potential compounds”. Please mention the 4 sites available for modification.

5.      Figure 3, all the 5 possible substituents need to be mention for better understanding.

6.      The methodology part needs to be readdressed for better understanding.

7.      The conclusion part is too long and need to be precise and reduced for better understanding and more focus need to be put on their findings.

8.      The article seems very lengthy in the current form; it is suggested to reduce the article by transferring some of the details as a separate supporting information file.

Author Response

Comments: The article entitled “Application of inverse design approaches to the discovery of nonlinear optical switches” is overall a good scientific effort that has been made by Mercedes Alonso et al. The background and purpose of this study is interesting and the research article is properly handled by the authors. I believe that this article fulfills the quality to be published in Journal of molecules and can be of interest for its readership after the suggested modifications.

Recommendation: Accept with minor revision

We thank the reviewer for their positive opinion on the scientific content of the article and we appreciate their recommendation for the acceptance of our work in the journal Molecules.

  1. The abstract need be precise and reduced for better understanding as in the current form it is too lengthy. 

Response 1: We have significantly shortened and rewritten the abstract. We hope this new version enhances the understanding on the content of our manuscript. The new abstract reads as follows:

Molecular switches, in which a stimulus induces a large and reversible change in molecular properties, are of significant interest in the domain of photonics. Thanks to their commutable redox states with distinct nonlinear optical (NLO) properties, hexaphyrins have emerged as a novel platform for multistate switches in nanoelectronics. In this study, we employ an inverse design algorithm to find functionalized 26R ® 28R redox switches with maximal bHRS contrast. We focus on the role of core-modifications as a synergistic effect with meso-substitutions was recently found for the 30R-based switch. In contrast to these findings, the inverse design optima and subsequent database analysis of the 26R-based switches confirm that core-modifications are generally not favored when high NLO contrasts are targeted. Moreover, while push-pull combinations enhance the NLO contrast for both redox-type switches, they prefer a different arrangement in terms of electron-donating and electron-withdrawing functional groups. Finally, we aimed at designing a three-state 26R ® 28R ® 30R switch having a similar NLO response for both ON states. Even though our best-performing three-state switch follows the design rules of the 30R-based component, our chemical compound space plots show that well-performing multistateswitches can be found in regions shared by high-responsive 26R and 30R structures.

  1. Keywords should be in the alphabetical order.

Response 2: We thank the referee for this suggestion and the keywords are now alphabetically ordered.

  1. The introduction part needs rephrasing….. and also need to be reduced

Response 3: We have significantly shortened and rewritten the introduction. We have changed slightly the structure of the introductionand remove several sentences and we think the readability on the introduction has significantly improved.

  1. Lines 132-133, “Taking the azoheteroarene scaffold as an example of a photoswitch scaffold (Figure 3) with 4 sites available for modification and 5 possible substituents, it leads to a chemical space region of 625 potential compounds”. Please mention the 4 sites available for modification.

Response 4: We have adapted Lines 132-133 to “Taking the azoheteroarene scaffold as an example of a photoswitch with 4 sets of sites (R1-R4) available for modification and 5 possible substituents, depicted as colored circles (Figure 3), it leads to a chemical space region of 625 potential compounds.” and we have added the same information in the caption of Figure 3. Moreover, we have included the labelling of the sites being optimized (e.g. site R1 optimization, site R2 optimization, …) in the upper part of Figure 3.

  1. Figure 3, all the 5 possible substituents need to be mention for better understanding.

Response 5: For a better understanding, we have included the labelling of the sites being optimized (e.g. site R1 optimization, site R2optimization, …) in the upper part of Figure 3. The 5 substituents are represented as colored circles, and we have included all this information in the caption of Figure 3.

The new caption of Figure 3 is as follows:

Figure 3. Schematic representation of the best-first search algorithm applied on an azoheteroarene
photoswitch with 4 modifiable (sets of) sites R1-R4 and a fragment library of 5 substituents, depicted
as blue, cyan, green, yellow and orange colored circles.

  1. The methodology part needs to be readdressed for better understanding. 

      Response 6: In line with the comments of Referee 2, we readdressed the methodology part by repositioning the equation on the βHRS to the methodology section so that the equation would appear earlier. In addition, we also explicitly state that the full tensor definition is used in the main text, and we refer to SI for the full tensor components.

“The first hyperpolarizability related to the Hyper-Rayleigh Scattering (βHRS) is an experimentally measurable property to probe the second-order NLO responses of the different states of the switches. [9,58–60] This property is related to the intensity of the incoherent scattered light at twice the frequency of the incident light (2ω). Under the condition that the laser’s propagation is perpendicular to the incoherent scattered light, the HRS equations based on the full tensor components, are written as Equation 1. The full tensor components can be found in the Supporting Information.”

In addition, we have clarified the information on how the BFS algorithm works and modified the Figures 3 and 4 to enhance the understanding of the methodology section.

  1. The conclusion part is too long and need to be precise and reduced for better understanding and more focus need to be put on their findings. 

Response 7: Regarding the conclusions, we have slightly reduced its length. In our humble opinion, the conclusion should contain the key results from the different sections of the article as well as highlight the novelty and significance of the work.

  1. The article seems very lengthy in the current form; it is suggested to reduce the article by transferring some of the details as a separate supporting information file.

Response 8: As both Referees stressed the need to shorten the current form of the paper, we transferred the part on the Computational Details to the Supporting Information. In addition, we shortened the abstract and the introduction, reduce considerably the number of references and consequently we have reduced substantially the length of the article.

Reviewer 2 Report

Comments and Suggestions for Authors

The present work by Desmedt et. al. is a computational study discussing optimization of molecular switches with high first hyperpolarizability contrast between their states. The authors use a site-by-site inverse design approach and density-functional theory calculations to identify high-performing switches and structure-property relationships. The work is well-described and uses proven DFT methods for calculation of hyperpolarizabilities. 

One outstanding question on the BFS algorithm used is its stability with respect to start point or permutation of order of sites in the search. A potential concern here would involve an overpowered donor or acceptor placing the system on the edge of a stable local region of high hyperpolarizability, where substituents at other locations cause a rapid decline and the algorithm gets locally stuck, where a smaller increase at the initial location may have allowed optimization to continue towards the global maximum. See for example Xu et al. J. Mat Chem. C. 2021, doi: 10.1039/d0tc05700b, in which this effect is discussed in the context of both donor and dielectric environment optimization.

Two other minor points:

While beta_HRS is defined in the manuscript, the definition should either appear earlier or the supporting information should be pointed to for the definition earlier in the paper. It should also be clarified in the manuscript that the full-tensor definition (as in the SI) is used and not a dominant-value approximation (as is often used for linear chromophores, beta_HRS ~ sqrt(6/35)*beta_zzz).

-Figure 11 could use clearer labeling to discuss the observed trends. 

Comments on the Quality of English Language

This manuscript requires significant English-language copyediting prior to publication, particularly in the introduction and abstract. While other sections of the paper had fewer syntax issues, more emphasis on avoiding run-on sentences and long paragraphs would assist with readability. A couple of notable issues:

1. I would recommend rewriting the first sentence of the abstract as "Molecular switches, in which a stimulus induces a large and reversible change in molecular properties, are of significant interest in the domain of photonics." (or similar)

2. In the first sentence of the supporting information, I assume the authors intended to write 'incoherent light' instead of 'inOHerent light.'

Author Response

Reviewer 2:

Comments and Suggestions for Authors

The present work by Desmedt et. al. is a computational study discussing optimization of molecular switches with high first hyperpolarizability contrast between their states. The authors use a site-by-site inverse design approach and density-functional theory calculations to identify high-performing switches and structure-property relationships. The work is well-described and uses proven DFT methods for calculation of hyperpolarizabilities.

One outstanding question on the BFS algorithm used is its stability with respect to start point or permutation of order of sites in the search. A potential concern here would involve an overpowered donor or acceptor placing the system on the edge of a stable local region of high hyperpolarizability, where substituents at other locations cause a rapid decline and the algorithm gets locally stuck, where a smaller increase at the initial location may have allowed optimization to continue towards the global maximum. See for example Xu et al. J. Mat Chem. C. 2021, doi: 10.1039/d0tc05700b, in which this effect is discussed in the context of both donor and dielectric environment optimization.

Response 1: We understand the reviewer’s concerns about the BFS algorithm getting trapped in local optima. We already mentioned this disadvantage in the original version of the manuscript. However, we did not report clearly enough how we usually address this potential issue. Our strategy to avoid obtaining a single, local optimum by exploring a too narrow region of CCS is to run multiple BFS procedures with starting points from very different regions in CCS. Also, important to mention is that we generally use random site orders. We therefore added the following sentences in the BFS methodology section:

“Nevertheless, also for BFS the issue of efficacy can be (partially) remedied. In this work, multiple BFS runs are conducted with a variance in start structure and site sequence. By selecting start structures from different regions of CCS, an exhaustive enough portion of CCS is being explored.”

As in our previous studies (Desmedt et. al. Front. Chem. 2021 doi:10.3389/fchem.2021.786036 and Desmedt et. al. Phys. Chem. Chem. Phys. 2023, doi: 10.1039/D3CP01240A), this strategy was also applied in this work. For the 26R  28R switch, four different runs were conducted with a focus on a change in core-modifications. All of them converged to the same optimal structure, in agreement with earlier findings (Desmedt et. al. Front. Chem. 2021). For the three-state switch, similarly four BFS procedures were executed, of which three of them yielded the most optimal switch found. We can therefore state with near certainty that in both cases one of the best possible switches was found using BFS. Finally, we wish to emphasize again that, regardless of the efficiency and even efficacy, the orderliness in collecting data during BFS substantially aids in extracting structure-property relationships and design rules.” 

In the last paragraph of subsection 3.1.1, we have added the following sentences:

“Note that three additional BFS runs using start structures for which a variation in type of core-modifiers is imposed, resulted in our aforementioned optimal structure. This makes it very plausible that we identified one of the most NLO efficient 26R-based switches.

Two other minor points:

-While beta_HRS is defined in the manuscript, the definition should either appear earlier or the supporting information should be pointed to for the definition earlier in the paper. It should also be clarified in the manuscript that the full-tensor definition (as in the SI) is used and not a dominant-value approximation (as is often used for linear chromophores, beta_HRS ~ sqrt(6/35)*beta_zzz).

Response 2:  We fully agree with the referee. We repositioned the equation of the βHRS to the methodology section so that the equation would appear earlier. In addition, we also explicitly state that the full tensor definition is used in the main text, and we refer to SI for the expressions of the full tensor components.

“The first hyperpolarizability related to the Hyper-Rayleigh Scattering (βHRS) is an experimentally measurable property to probe the second-order NLO responses of the different states of the switches. [9,58–60] This property is related to the intensity of the incoherent scattered light at twice the frequency of the incident light (2ω). Under the condition that the laser’s propagation is perpendicular to the incoherent scattered light, the HRS equations based on the full tensor components, are written as Equation 1. The full tensor components can be found in the Supporting Information.”

-Figure 11 could use clearer labeling to discuss the observed trends.

Response 3:  We have clarified the labeling in Figure 11 by adding a legend explaining the colors defining the different regions.

Comments on the Quality of English Language

This manuscript requires significant English-language copyediting prior to publication, particularly in the introduction and abstract. While other sections of the paper had fewer syntax issues, more emphasis on avoiding run-on sentences and long paragraphs would assist with readability.

Response 4:  As suggested by another reviewer, we have rewritten and shortened the abstract and introduction considerably. We hope that the current versions meet the English-language standards.

A couple of notable issues:

  1. I would recommend rewriting the first sentence of the abstract as "Molecular switches, in which a stimulus induces a large and reversible change in molecular properties, are of significant interest in the domain of photonics." (or similar)

Response 5:  We thank the reviewer for this suggestion and included it into the first sentence of the revised abstract.

  1. In the first sentence of the supporting information, I assume the authors intended to write 'incoherent light' instead of 'inOHerent light.'

Response 6:  We thank the reviewer to point out this spelling mistake; we have corrected it.
